# Botulinum Toxin Treatment of Adult Muscle Stem Cells from Children with Cerebral Palsy and hiPSC-Derived Neuromuscular Junctions

**DOI:** 10.3390/cells12162072

**Published:** 2023-08-15

**Authors:** Domiziana Costamagna, Valeria Bastianini, Marlies Corvelyn, Robin Duelen, Jorieke Deschrevel, Nathalie De Beukelaer, Hannah De Houwer, Maurilio Sampaolesi, Ghislaine Gayan-Ramirez, Anja Van Campenhout, Kaat Desloovere

**Affiliations:** 1Neurorehabilitation Group, Department of Rehabilitation Sciences, KU Leuven, 3000 Leuven, Belgium; domiziana.costamagna@kuleuven.be (D.C.); valeria.bastianini@edu.unito.it (V.B.); nathalie.debeukelaer@kuleuven.be (N.D.B.); 2Stem Cell and Developmental Biology Unit, Department of Development and Regeneration, KU Leuven, 3000 Leuven, Belgium; marlies.corvelyn@kuleuven.be (M.C.); robin.duelen@kuleuven.be (R.D.); maurilio.sampaolesi@kuleuven.be (M.S.); 3Cardiology, Department of Cardiovascular Sciences, KU Leuven, 3000 Leuven, Belgium; 4Laboratory of Respiratory Diseases and Thoracic Surgery, Department of Chronic Diseases and Metabolism, KU Leuven, 3000 Leuven, Belgium; jorieke.deschrevel@kuleuven.be (J.D.); ghislaine.gayan-ramirez@kuleuven.be (G.G.-R.); 5Willy Taillard Laboratory of Kinesiology, Geneva University Hospitals and University of Geneva, 1211 Geneva, Switzerland; 6Department of Orthopedic Surgery, University Hospitals Leuven, 3000 Leuven, Belgium; hannah.dehouwer@uzleuven.be (H.D.H.); anja.vancampenhout@uzleuven.be (A.V.C.); 7Department of Development and Regeneration, KU Leuven, 3000 Leuven, Belgium

**Keywords:** cerebral palsy in young children, BoNT cell treatment, muscle microbiopsy, adult muscle stem cells, induced pluripotent stem cells, myogenesis, collagen

## Abstract

Botulinum neurotoxin type-A (BoNT) injections are commonly used as spasticity treatment in cerebral palsy (CP). Despite improved clinical outcomes, concerns regarding harmful effects on muscle morphology have been raised, and the BoNT effect on muscle stem cells remains not well defined. This study aims at clarifying the impact of BoNT on growing muscles (1) by analyzing the in vitro effect of BoNT on satellite cell (SC)-derived myoblasts and fibroblasts obtained from medial gastrocnemius microbiopsies collected in young BoNT-naïve children (t0) compared to age ranged typically developing children; (2) by following the effect of in vivo BoNT administration on these cells obtained from the same children with CP at 3 (t1) and 6 (t2) months post BoNT; (3) by determining the direct effect of a single and repeated in vitro BoNT treatment on neuromuscular junctions (NMJs) differentiated from hiPSCs. In vitro BoNT did not affect myogenic differentiation or collagen production. The fusion index significantly decreased in CP at t2 compared to t0. In NMJ cocultures, BoNT treatment caused axonal swelling and fragmentation. Repeated treatments impaired the autophagic–lysosomal system. Further studies are warranted to understand the long-term and collateral effects of BoNT in the muscles of children with CP.

## 1. Introduction

Cerebral palsy (CP) is the single largest cause of childhood physical disability, with a prevalence of 2 to 3 per 1000 live births [1], and is characterized by a non-progressive brain lesion. The brain injury causes neuromuscular alterations that originate in early childhood [2], progressing during life [3]. Many reports have recognized muscle alterations, such as muscle atrophy, altered fiber distribution and a reduced number of satellite cells (SCs), together with a hypertrophic extracellular matrix [4,5,6,7,8,9,10]. However, the experimental design of the majority of these studies did not always foresee a correct comparison between muscle, age and reference group. Therefore, there is an urgent need for specific muscle-dedicated studies with appropriate matched controls.

The most common CP phenotype is characterized by spasticity, which is observed in 60–85% of children in developed countries [11]. The first line of treatment to correct typical focal muscle hyperactivity in children with CP is represented by local botulinum neurotoxin type-A (BoNT) injections. BoNT is available in seven isoforms of bacterial origin and is mainly responsible for the blockage of the presynaptic release of acetyl choline, the neuromuscular transmitter that gives the muscle the input to contract. In particular, the endoproteolytic activity of BoNT cleaves SNAP25, one of the components of the SNARE complex (soluble-N-ethylmaleimide-sensitive factor attachment protein receptor), the synaptic vesicle-docking fusion complex at the level of neuromuscular junctions (NMJs). Cleavage of this complex abrogates vesicle fusion and blocks the transmission of acetyl choline from the synapse, causing paralysis of the muscle [12].

The blockage of muscle contraction lasts from three to six months and MRI analysis has reported gastrocnemius alterations in healthy volunteers after a single BoNT administration compared to the contralateral saline-injected muscle. Moreover, muscle atrophy, different fiber type distribution and bimodal distribution of fiber dimension, and increases in the number of perimysial fat cells and the endomysial connective tissue around atrophic fibers were reported 12 months after incobotulinum toxin A (Xeomin) administration [13,14]. These, together with an increase in non-contractile tissue and fat accumulation in BoNT-treated animal models, have raised serious concerns. However, the results in animal studies are not comparable or directly translatable to patients due to the use of a higher dose and shorter interval of time between one or repetitive injections [15,16,17].

In children with CP, BoNT treatment has been shown to decrease muscle tone and ameliorate mobility and muscle function, eventually resulting in gait improvement [18]. Moreover, the possibility of applying BoNT treatment at a multidimensional level is encouraging, exploiting its antinociceptive role, counteracting neuropathic pain and improving patient quality of life [19]. While the short-term positive effects of BoNT on muscle functionality have largely been reported [20,21,22,23], risks of detrimental (long-term) effects of BoNT on muscle growth and contractility have been highlighted, further compromising muscle weakness [24,25,26,27,28] or inducing a further worsening of muscles that have already been subjected to reduced growth [29]. These recent indications seem to motivate clinicians and caretakers to use other treatments, usually in combination (such as physiotherapy or serial casting) instead of simply relying on BoNT [30]. Nevertheless, there is still space for improvement, since the mechanism through which BoNT causes direct or indirect intrinsic muscle alterations and weakness remains unknown. Additionally, BoNT is commonly used to ameliorate spasticity in stroke patients. Direct BoNT treatment of adult stroke patient-derived myoblasts showed a significantly increased proliferative phenotype and impaired myogenic differentiation to myotubes by decreasing the expression of contractile genes. Moreover, BoNT promoted the expression of proinflammatory genes in the muscle-resident fibroblasts, which are mainly responsible for fibrotic tissue and extracellular matrix deposition [31]. Still, it is not known whether BoNT could induce similar alterations in SC-derived myoblasts, which are reasonably responsible for muscle growth in young children and are the effective precursors that will become activated after muscle damage. Similarly, it is currently unknown whether BoNT could stimulate the production of fibrotic tissue, which could alter the proportion of contractile tissue in young children with CP and negatively impact muscle recovery, structure and function. How these features are developing and the way the muscle cells react to BoNT over time are still unknown. A recent publication demonstrated the inhibitory effect of BoNT on myotube contraction when administered to motor neurons and myotube cocultures [32], although the specific consequences for both muscles and neurons remain to be fully elucidated.

Therefore, the goal of this study is to clarify the effect of BoNT on growing muscles with CP (1) by determining the direct effect of in vitro BoNT treatment on SC-derived myoblasts and fibroblasts obtained from medial gastrocnemius (MG) microbiopsies collected in young BoNT-naïve CP children (time 0, t0) and compared to typically developing (TD) children with healthy growth development in the same age range; (2) by following the effects related to post-BoNT in vivo administration to the same children with CP on the cells obtained from MG 3 (t1) and 6 months (t2) after the first BoNT injection over time. We hypothesize that direct in vitro BoNT treatment will impact myogenic differentiation and collagen synthesis and worsen the expression of fibrotic markers in a time-dependent manner. Finally, the direct effect of single and repeated in vitro administration of BoNT on both muscle and neural compartments was examined in a microfluidic NMJ model, consisting of a coculture of human induced pluripotent stem cell (hiPSC)-derived lower motor neurons (MNs) and autologous hiPSC-derived myotubes. We hypothesize that BoNT will affect NMJ morphology by altering the binding of lower MNs to the myotubes and increase muscle proteolysis.

The results obtained in this study demonstrated the ability of BoNT to not induce major alterations of muscle stem cells when applied in vitro but to cause a decrease in FI from SC-derived myoblasts extracted from children with CP 6 months after the first in vivo BoNT administration. Unexpectedly, no fibroblast-dependent collagen deposition was observed. Finally, the more comprehensive NMJ model after BoNT treatment showed neural damage and disruption of NMJ connections on one side, with activation of muscle proteolysis through the ubiquitin–proteasome pathway (Atrogin1 increase) on the other one. Moreover, repetitive BoNT treatments induced a blockage in the autophagic–lysosomal system (p62 accumulation).

## 2. Materials and Methods

### 2.1. Recruitment

Patient recruitment was performed between February 2020 and April 2022. The study protocol was authorized by the Ethical Committee of the University Hospitals Leuven (Belgium; S62645). The parents or legal guardians of all participants signed the written informed consent. Children with CP were enrolled from the CP Reference Centre, where approximately 3830 contacts with children with CP in the multidisciplinary CP clinic were screened. Children with CP recruited for this study (a) had a diagnosis of spastic CP confirmed by a neuropediatrician; (b) were aged between 2 and 9 years; (c) were classified as GMFCS level I, II or III; (d) had no previous BoNT injection; (e) had no history of orthopedic or neurological surgery; and (f) were clinically planned for their first BoNT treatment in the medial gastrocnemius (MG) muscle. This resulted in a BoNT-naïve treatment group (i.e., no previous history of BoNT injection). CP children with the presence of dystonia or ataxia, orthopedic surgery less than 2 years before or any surgery on the included MG muscle were excluded. All patients wore orthoses and underwent regular physiotherapy as part of their standard of care. TD children were recruited from the Traumatology Unit (on average, 70 children during the time period of the study) or the Ear-Nose-Throat Unit of the University Hospital of Leuven (Belgium; on average, 85 children during the period of the study). TD children were selected based on the following criteria: (a) the same ultrasound settings were used to acquire TD and data from children with CP and (b) children were aged between 2 and 9 years, in accordance with the age distribution of all children in the BoNT treatment groups. TD children were further excluded when they presented a history of neurological or metabolic problems or when following a high-performance sporting program. During the study period, a total of 268 children with CP and 30 TD children met the selection criteria. The power analysis used to define the study sample was performed based on previous cell cycle percentages reported by Zanotti and co-workers [31], who defined the effect of in vitro BoNT treatment on quadricep myoblasts extracted from adult stroke patients. Based on their effect size of 1.44 for the comparison of cell cycle outcomes pre- and post-BoNT treatment, a sample size of 6 children per study group was needed for the current study (paired *t*-test; two-tailed; power: 0.80; α of 0.05; G Power version 3.1.9.7). To take risk of dropout into account, based also on previous studies [33,34], the sample size was slightly increased (CP n = 10; TD n = 7). A final group of 10 children with uni- or bilateral CP (age 4.11 ± 1.66; range: 3.0–9.0 years) with gross motor function classification system (GMFCS) level I, II or III and 7 TD children in the same age range (age 6.19 ± 1.92; range: 3.0–9.0 years) were included in this study. A flow diagram of patient recruitment and selection is presented in Appendix A.

The MG was selected for BoNT injections, and the total dose per muscle was decided by a pediatric orthopedic specialist after clinical examination and 3D gait analysis that routinely took place just before each BoNT treatment. The ultrasound-guided BoNT (Botox^®^, Allergan, Irvine, CA, USA) injection was dispensed in the MG muscle in a day-clinic setting at a concentration of 20 IU/mL under general anesthesia. In general, the dosage, depending on the targeted MG muscle, the spasticity degree and its interference with the function of the patient, averaged between 2 and 4 IU/Kg of body weight [35].

### 2.2. Biopsy Collection

MG muscle microbiopsies from the muscle belly were obtained during interventions under general anesthesia (at the time of the first BoNT injection (t0) or during orthopedic surgery). In TD children, microbiopsies were taken when osteosynthesis material from upper limb trauma surgery was removed or when the child was undergoing ear–nose–throat surgery. Muscle microbiopsies of the MG muscle of the same young CP patients 3 months (t1) and 6 months (t2) post BoNT were obtained under local sedation with Kalinox [36,37,38]. Ultrasound-guided percutaneous biopsy collections were performed using a microbiopsy needle (16-gauge, Bard). The acquired microbiopsies were 1.5 cm long, with a diameter of 1.3 mm and a weight of less than 10 mg (fresh biopsy weight, TD: 8.0 ± 0.27 mg; CP: 8.5 ± 0.46 mg; *p* = 0.5714; see also Appendix A). During the first days after the procedure, the clinical tolerance was good, and parents evaluated the associated pain. In the first two days after biopsy collection, a few subjects experienced some discomfort, stiffness and minor pain. Later on, no complaint was reported by any of the subjects.

### 2.3. Cell Culture

Muscle biopsies were disrupted in smaller pieces and cultured in dishes precoated with bovine collagen (0.1%; Merck, KGaA, Darmstadt, Germany) at 37 °C in a 5% CO_2_ atmosphere according to the laboratory transplant technique [34]. Growth medium based on IMDM (Thermo Fisher Scientific, Waltham, MA, USA, for all the compounds, as differently specified) was supplemented with 20% fetal bovine serum (FBS), 1 mg/mL non-essential amino acids, 1 mg/mL sodium pyruvate, 100 IU/mL penicillin/streptomycin (P/S), 1% chicken embryo extract (CEE; Bio Connect, The Netherlands), 2 mM glutamine and 100 nM β-mercaptoethanol. Cells were passaged upon reaching 70% confluence using TrypLE^TM^ Express or prepared for cell sorting.

### 2.4. Fluorescence-Activated Cell Sorting

Different cell populations were isolated by serial FACS with an MA900 Multi-Application Cell Sorter (Sony Biotechnology, San Jose, CA, USA) under a BSL2 microbiological safety hood. Cells were incubated in the dark with the primary antibodies for 30 min at room temperature (RT). Bare cells and negative controls (fluorescence minus one) were applied to assess the correct gating. A fraction of cells was used to determine the viability gate, using 10 μM calcein violet, a staining for the viable cells that is metabolized by intracellular esterases emitting a signal in the green fluorescence spectrum. Dead cell gating was determined by heat burning a small aliquot of cells for 15 min at 65 °C. For the first FACS, the total population of cells protruding from the microbiopsy and further amplified (three to ten million cells) was sorted at a low passage (p2 or p3) for a recognized marker for SC-derived myoblasts (CD56) [39,40]. After this first sort, cells were further amplified, and the CD56^−^ population was used for the second FACS a couple of passages later (p4 ± 1). By applying a mesoangioblast marker (MAB; Alkaline Phosphatase, ALP) and a fibro-adipogenic progenitor cell marker (FAPs; platelet-derived growth factor receptor alpha, PDGFRa) as reported in Table 1, the double-negative population (a fraction of the CD56^−^ ALP^−^ PDGFRa^−^ population) was sorted out as a population mainly enriched in fibroblasts. Further negative markers were used to test the absence of contaminating cells, such as CD45, CD31, CD34 and CD117 [41]. All cell populations were collected and amplified for further experiments. After sorting, the cells were seeded at a minimum density of 1.7 × 10^3^ cells/cm^2^ and amplified for further experiments.

### 2.5. In Vitro BoNT Treatment

BoNT treatment was applied by directly reconstituting BoNT (Allergan) in the growth medium at a 20 IU/mL concentration, which is the usual concentration used in clinical treatment [24,35] and previously applied under similar conditions on biopsy-derived muscle cells and fibroblasts [31].

SCs were seeded at 16 × 10^3^ cells/cm^2^. Then, 24 h after seeding, the growth medium was changed to growth medium supplemented with 20 IU/mL BoNT, while control cells received fresh untreated medium. After 48 h treatment, both control and BoNT-treated cells received untreated differentiation medium consisting of high-glucose DMEM, 2% horse serum (HS), 1% CEE, 1 mg/mL sodium pyruvate and 100 IU/mL P/S. The medium was replaced every other day, and myogenic differentiation was terminated after 6 days for all samples (Appendix A).

Fibroblasts were seeded at 25 × 10^3^ cells/cm^2^ in multiwell dishes with growth medium. Then, 24 h after seeding, the growth medium was changed to growth medium supplemented with 20 IU/mL BoNT, while control cells received new untreated medium. After 48 h of treatment, fibroblasts were detached, pelleted and frozen for further analyses (Appendix A).

### 2.6. Human Induced Pluripotent Stem Cell Culture

Healthy donor hiPSC lines were cultured on dishes coated with Geltrex LDEV-Free hESC-Qualified Reduced Growth Factor Basement Membrane Matrix (Corning, USA). Cells were maintained in Essential 8 Flex Basal Medium complemented with Essential 8 Flex Supplement (50×) and 10 IU/mL P/S. Cultures were maintained at 37 °C under normoxic conditions (21% O_2_ and 5% CO_2_). The colonies were passaged non-enzymatically with 0.5 mM EDTA in phosphate-buffered saline (PBS). The following hiPSC control lines were used, as previously reported [42,43]: HC #1 (A18945, Thermo Fisher Scientific) derived from female CD34^+^ cord blood, applying a seven-factor EBNA-based episomal system; HC #2 supplied by Prof. C. Verfaillie (University of Leuven, Belgium) and obtained after lentiviral transduction of the BJ1 fibroblast cell line of a new-born male subject, as detailed elsewhere [44]; and HC #3 generated by Sendai virus-based reprogrammed male donor fibroblasts (SBAD2), which was a gift to Prof. C. Verfaillie from Prof. P. Jennings (Medizinische Universität, Innsbruck, Austria), as previously published [45]. Periodically, dedicated assays were conducted to assess the presence of mycoplasma contamination.

### 2.7. Skeletal Muscle Differentiation

Following a skeletal muscle differentiation protocol that was previously described in [43], with some minor adaptations, hiPSCs were differentiated into myocytes. Cells were incubated with 0.2% Rho-associated protein kinase (ROCK) inhibitor (Y-27632; Merck; Darmstadt, Germany) and detached with Accutase Solution (Merck) as single cells at every cell passage and during the first day of differentiation. The mesoderm stage (day 0) was reached after seeding 5000 cells/cm^2^ on Matrigel Growth Factor Reduced (GFR) Basement Membrane Matrix layer (Corning) in the induction medium for skeletal muscle (M1) for 4 to 6 days. This medium was composed of Essential 6 Medium supplemented with 5% HS, 3 μM CHIR99021 (Axon Medchem, Reston, VA, USA), 2 μM SB431542 (Merck), 10 ng/mL human recombinant Epidermal Growth Factor (EGF; PeproTech; London, UK) and 0.4 μg/mL water-soluble dexamethasone (Merck). When the cells reached 90–100% confluency, they were dissociated and reseeded at the initial density of 5000 cells/cm^2^ in skeletal myoblast medium (M2). M2 was composed of Essential 6 Medium supplemented with 5% HS, 10 ng/mL EGF, 20 ng/mL recombinant human (insect-derived) hepatocyte growth factor (HGF), 10 ng/mL recombinant human platelet-derived growth factor BB (PDGF-BB), 20 ng/mL human fibroblast growth factor-basic (FGF-2), 20 μg/mL oncostatin, 10 ng/mL insulin-like growth factor 1 (IGF-1; all from PeproTech) and 0.4 μg/mL dexamethasone. After reaching 90–100% confluency, 6 to 8 days later, myocytes were trypsinized to be frozen or to induce differentiation into myotubes during the following 7 days. Differentiation medium (M3), was composed of Essential 6 Medium complemented with 50 nM necrosulfonamide (R&D Systems, Minneapolis, MN, USA), 20 μg/mL oncostatin and 10 IU/mL P/S. The M3 was changed every 2 days.

### 2.8. Lower Motor Neuron Differentiation

A fast and efficient differentiation protocol previous published [43], with minor adaptations, was used to induce hiPSC differentiation to lower MNs. Briefly, 1 h pretreatment with 0.2% ROCK inhibitor was applied while detaching the cells with Accutase Solution as single cells and during the first day of differentiation. Then, 0.5 × 10^5^ cells/cm^2^ cells were seeded on 1 h GFR-coated plates in neural induction medium (N1). This medium consists of a 1:1 mix of KO-DMEM/F12 and neurobasal medium (NBM) complemented with 10% knockout serum replacement (KOSR), 2 mM glutamine, 0.1 mM ascorbic acid 2-phosphate (AA; Merck), 1% non-essential amino acids (NEAAs), 3 μM CHIR99021 (Axon Medchem), 2 μM SB431542, 1 μM compound E and 1 μM dorsomorphin (the last three from Merck). Every day, the medium was changed. After dissociation, 7 days later, cells were plated in expansion medium (N2) composed of a 1:1 mixture of KO-DMEM/F12 and NBM medium composed of 1% NEAA, 1% glutamine, 1% B27, 1% N2, 10 ng/mL FGF-2, 0.1 mM AA and 10 ng/mL EGF. After reaching confluence, cells were frozen for further analyses or additionally differentiated to reach the stage of neural precursor cells. These cells were obtained in MN induction medium (N3) composed of a 1:1 mixture of KO-DMEM/F12 and NBM with 1% NEAA, 1 mM glutamine, 1% B27, 1% N2, 0.1 mM AA, 100 ng/mL recombinant human Sonic Hedgehog (SHH; PeproTech), 10 μM all-trans-retinoic acid (Merck), 1 mM Smoothened Agonist (SAG) dihydrochloride and 1 μM purmorphamine (the last two from Merck). Finally, after 7 days, the cells were detached and replated in lower MN maturation medium (N4) composed of a 1:1 mixture of KO-DMEM/F12 and NBM complemented with 1% NEAA, 1 mM glutamine, 1% B27, 1% N2, 10 IU/mL P/S, 0.1 mM AA, 10 ng/mL recombinant brain-derived neurotrophic factor (BDNF), 10 ng/mL recombinant human ciliary neurotrophic factor (CNTF), 10 ng/mL recombinant human glial-derived neurotrophic factor (GDNF) and 10 ng/mL recombinant human neurotrophin 3 (NT-3; all from PeproTech). Lower MNs were maintained in this medium, and the medium was changed every 2 days.

### 2.9. Neuromuscular Junction Formation Using Microfluidic Devices

Microfluidic devices favor the growth of different cell types in a compartmentalized way to allow for the interaction of MNs and myotubes and in the end-form NMJs [43,46]. XonaChips devices (XC150; Xona Microfluidics; NC, USA) were applied as suggested by the producer with small cell- and culture-specific adaptations. A GFR Matrigel coating was applied before seeding differentiated hiPSC-derived myoblasts and autologous lower MNs. First, 0.1 × 10^5^ MNs were plated in one of the two compartments present in the middle channel. One day later, axons were attracted to the opposite compartment by applying N4 medium supplemented with the following four neurotrophic factors: 40 ng/mL CNTF, 40 ng/mL BDNF, 40 ng/mL GDNF and 40 ng/mL NT-3. The soma of the lower MNs was maintained in N4 medium that was neurotrophic factor-deprived. After 7 days, in the opposite channel, 7 × 10^3^ myoblasts were seeded and maintained in M2 medium. The following day, myotube differentiation was induced in the muscle compartment by replacing M2 medium with M3 medium. After 24 h, M3 medium was mixed with N4 medium in a 1:1 ration and supplemented with neurotrophic factors. Microfluidic cultures, giving rise to colocalization of neurofilament heavy polypeptide (NEFH) with Bungarotoxin (Bgt)-positive clusters of nicotinic acetyl choline receptors (AChRs) in sarcomeric α-actinin (ACTN2)-positive areas, were kept for 11 days, with the medium changed every day.

### 2.10. Coculture Experiments and Neuromuscular Junction Botulinum Toxin Treatment

NMJ coculture experiments were performed as previously published [43]. On a layer of myotubes (day 1 of myotube differentiation), 1 × 10^4^ cells/cm^2^ lower MNs were plated to allow for lower MN attachment and NMJ development. Cocultures were kept in a 1:1 mixture of M3 medium and N4 medium for 11 days. The medium ensured complete myotube differentiation. Both NMJ microfluidics in cocultures were treated for 48 h with a 1:1 mixture of M3 medium and N4 medium supplemented with 20 IU/mL BoNT, while control cocultures received a simple, untreated 1:1 mixture of M3 medium and N4 medium. The time interval for BoNT treatment (2 days) and recovery (2 days) was chosen based on previous available literature [31,47,48].

In particular, for WB analyses, experiments on NMJs were conducted in a coculture of six well plates in order to obtain enough protein lysate to test the expression levels of different proteins.

First, to investigate the direct effect of BoNT treatment on the target and to validate our model, a time course of 1 to 5 days was performed, sampling cells each day over time (for a better visualization, see Appendix A). Cocultures were prepared by seeding 10,000 myocytes/cm^2^ in proliferating medium (M2) for 2 days, followed by differentiation towards myotubes the day after with M3 medium. On day 0, 15,000/cm^2^ lower MNs were seeded on top of myotubes in N4 medium to favor their attachment and proliferation. From this point onwards, a 1:1 mixture of M3 medium and N4 medium was applied, and the medium was changed every other day. Treatment with 20 IU/mL BoNT (dissolved in a 1:1 mixture of M3 and N4 media) started on day 7 and lasted until day 11. These experiments were performed with the aim of validating the model by confirming proteolysis activation with BoNT treatment in hiPSC-derived NMJs, as already largely demonstrated in animal experiments [49,50,51].

Secondly, to mimic the situation of muscles treated multiple times, repetitive BoNT treatments were performed to molecularly follow the changing of cocultures repetitively stressed by BoNT treatments. Therefore, cocultures were treated for 48 h with 20 IU/mL BoNT, followed by 48 h recovery in normal medium and compared to untreated cocultures. This procedure was applied twice (see Appendix A for a better visualization of the experiment). For these consecutive treatments, the experiment started on day 3 in 6-well formats, where 48 h 20 IU/mL BoNT treatment was applied in four wells, while untreated wells received a 1:1 mixture of M3 and N4 media (Untr). On day 5, cells in one well were collected (indicated as 1× BoNT), and the cells in the remaining five wells were kept for another 2 days in a normal 1:1 mixture of M3 and N4 media (recovery period, Rec). At this point, after collecting 1× untreated and 1× BoNT + Rec wells (day 7), the second treatment was repeated for 48 h with 20 IU/mL BoNT on two of the remaining wells. After this (day 9), collection of one well (2× BoNT) and a further 48 h with a normal 1:1 mixture of M3 and N4 media (‘2× Untr’ and ‘2× BoNt + Rec’) were performed. At the end of the experiment (day 11), the remaining two wells (indicated as ‘2× Untr’ and ‘2× BoNT + Rec’) were pelleted. The experiment was repeated with the three available cell lines, always creating cocultures with lower MNs and myotubes and treating the four wells in the same way.

### 2.11. Quantitative Real-Time PCR Analysis

Quantitative real-time PCR (qRT-PCR) was used to analyze the induction of genes expressed in fibrotic tissue by checking collagen genes in muscle fibroblasts present in the microbiopsies. RNA extraction from fibroblasts was performed with a PureLink RNA Mini Kit, and traces of genomic DNA were cleared with a TURBO DNA-Free DNase Kit following the supplier instructions. To reverse transcribe 500 μg of RNA, a SuperScript III Reverse Transcriptase First-Stranded Synthesis SuperMix was used. The resulting cDNA was dispensed in 384 wells pre-filled with Platinum SYBR Green qPCR SuperMix-UDG and 250 nM primers. qRT-PCR (40 cycles; 95 °C, 15 s; 60 °C, 45 s) was run and analyzed on a ViiA 7 Real-Time PCR plate reader. Different collagen genes implicated in collagen synthesis and deposition (all from IDT) were tested, as reported in Table 2. By subtracting the Ct values from the genes of interest with the Ct values of the housekeeping gene (*ACTB*), the Delta Ct (DCt) values were obtained.

### 2.12. Immunofluorescent Staining

To visualize SC-derived myotubes or myotubes and lower MNs forming NMJs, immunofluorescent staining (IF) was performed. Both SC-derived myotube cultures and NMJ cocultures were fixed with 4% paraformaldehyde (PFA; Polysciences; Warrington, PA, USA) and permeabilized with 0.2% Triton X-100 in 1% (*w*/*v*) bovine serum albumin (BSA; both from Merck). Cultures were blocked with a solution containing donkey serum (10% dilution in PBS; VWR; The Netherlands). Primary antibodies important for both cytosolic and nuclear expression of accepted markers indicating structure and expression (reported in Table 1) were incubated overnight at 4 °C in PBS and supplemented with 1% BSA. Secondary Alexa Fluor antibodies were diluted at 4 μg/mL in PBS supplemented with 1% BSA for 1 h at RT. To localize NMJs, Alexa Fluor 647-conjugated α-Bgt was used to stain the clusters of AChRs (red), colocalizing with neurofilament H (green) on the ACTN2^+^ area (grey) of the sarcomeres in the myotubes. Nuclei were counterstained with 10 μg/mL Hoechst (33342; blue), and FluorSave reagent was used as a mounting medium. For the quantification of SC-derived myotubes, five random images per condition were analyzed, for a total of 10 pictures per subject. NMJ buttons appeared as either rudimentary structures (circular contact points) or elongated, spreading structures (with broad and parallel multiple contact points) [43,52]. For the quantification of NMJs, an average of more than 60 pictures per condition were collected for each cell line using a Leica DMi8 inverted microscope with LASX software (LASX Office 1.4.5, Leica Microsystems, Wetzlar, Germany). Focal enlargement of specific areas of the neurites/axon equal to twice the diameter of the neurites/axon was reported as axonal swelling, and the per picture was counted, as previously reported [43,53,54,55,56,57,58]. NMJ fragments characterized by an interrupted fragment of a neurite not connected to other structures on either side were observed in each picture, as previously reported [59,60,61,62].

Western blot (WB) analysis was performed to quantify specific protein expression on lysates from fibroblasts, lower MNs or NMJ cocultures in order to prove the modulation of important markers of expression, differentiation or functionality. Samples were sonicated in RIPA buffer complemented with 0.5 mM sodium orthovanadate, 10 mM sodium fluoride, 1 mM phenylmethylsulfonyl fluoride and 1:100 protease inhibitor cocktail (all from Merck). Equal volumes of protein (25 µg) were heat-denatured with sample-loading buffer composed of 50 mM Tris-HCl, pH 6.8, 100 mM DTT, 2% SDS, 0.1% bromophenol blue and 10% glycerol. The proteins were resolved by SDS-polyacrylamide gel electrophoresis and transferred to nitrocellulose membranes (Amersham Protran Western Blotting Membranes; Merck). Tris-buffered saline (TBS) containing 5% non-fat dry milk (Merck) and 0.05% Tween was used to saturate the membranes for 1 h at RT, followed by overnight incubation with the specific primary antibodies (see Table 1). The secondary horseradish peroxidase (HRP)-conjugated antibodies (Bio-Rad; Hercules, CA, USA) were diluted 1:5000 in 0.05% TBS-Tween and 2.5% non-fat dry milk (Merck). After washing, SuperSignal Pico or a Femto chemiluminescence substrate was used to incubate the membranes and show the protein bands detected with the GelDoc Chemiluminescence Detection System (Bio-Rad). Using QuantityOne software 4.6.6 (Bio-Rad), quantification of relative densitometry was performed by subtracting the background signal from the protein band signal and the appropriate loading protein. An insufficient amount of samples was collected for GMFCS level III patients. Therefore, no analysis of collagens and fibronectin expression was performed for this group of patients. After COL1A1 and COL3A1 expression levels were presented, the ratio of the two proteins was available, as a decrease in COL1A1 usually corresponds to an increase in COL3A1 in the case of rearrangements [63,64].

### 2.13. Statistics

The data reported in this study show a normal distribution according to a Shapiro–Wilk test. All data are indicated as mean ± standard deviation (SD), except for gene expressions (mean ± standard error of the mean; SEM). Student’s *t*-test was used when two groups were compared. One-way or two-way ANOVA (with multiple post hoc comparisons test and Tukey or Sidak corrections, respectively) were used when analyzing three or more groups. Statistical tests were performed using Prism software 8.4.0 (GraphPad, San Diego, CA, USA). The significance of the differences is indicated as follows: * *p* < 0.05; ** *p* < 0.01; *** *p* < 0.001 and **** *p* < 0.0001 vs. TD; $ *p* < 0.05; $$ *p* < 0.01 vs. CP t0.

## 3. Results

### 3.1. Participants

A group of 10 children with uni- or bilateral CP (age: 4.11 ± 1.66; range: 3.0–9.0 years) with GMFCS level I, II or III and 7 TD children in the same age range (age: 6.19 ± 1.92; range: 3.0–9.0 years) were included in the study. Demographic and anthropometric characteristics of the enrolled children are presented in Table 3.

### 3.2. Reduced Myotube Differentiation after In Vivo BoNT Administration

To understand whether BoNT may have an in vitro effect on myogenic differentiation of cells extracted from CP patients, a comparison between untreated and BoNT-treated SC-derived myoblasts was performed after myogenic differentiation (see also Appendix A). Immunofluorescent images (IF) did not show significant differences in myotube conformation, either between CP and TD or after in vitro BoNT treatment, presenting no differences in myotubular structures stained with cytoplasmic myosin heavy chain (MYHC; red) and in nuclear expression for myogenic transcription factor MYOD (green; Figure 1A,B). After quantification (Figure 1C), no significant differences were reported between TD and CP FI (round symbols, representing the average of five fields of view for each subject-derived cell line), nor for the corresponding BoNT-treated cells (squared symbols, representing the average of five fields of view for each subject-derived treated cell line), confirming the qualitative IF data. The proportion of MYOD^+^ nuclei among the total number of nuclei, which is indicative of cells still able to undergo the myogenic program, did not differ significantly relative to the BoNT treatment (Figure 1D). In addition, there were no significant differences in the total number of nuclei (Figure 1E), both when comparing CP versus TD data and treated versus untreated cells. After quantification, no significant differences in FI between treated and untreated cells were observed (Figure 1H). However, for CP cells, a significantly lower FI was reported at t2 (6 months after in vivo BoNT administration) compared to t0 (t0 versus t2, *p* = 0.045; Figure 1I). Some enlarged MYHC^+^ areas could be observed in IF images of SC-derived myotubes extracted from children with CP at t1 (3 months after in vivo BoNT administration) and t2, with accumulation of nuclei not spreading along the myotube but remaining in one central area, independent of in vitro BoNT treatment (representative examples in Figure 1F,G), as previously reported [34]. In contrast, the fraction of MYOD^+^ nuclei after differentiation and the total number of nuclei did not show any significant difference after in vivo BoNT administration among the time points (Figure 1J; Appendix A). Nevertheless, the weight of the biopsies for cell isolation was the same between CP and TD samples (Appendix A).

Taken together, these results confirm no significant alterations attributable to the 48 h in vitro BoNT treatment for FI, MYOD^+^ cells and the total number of nuclei. However, BoNT injection results in a decreased FI value from cells at t2.

### 3.3. Collagen Expression Pattern of Fibroblasts after In Vitro BoNT Treatment

Fibroblasts are important for muscle homeostasis and are responsible for the secretion of extracellular matrix proteins such as collagens under stress conditions [65]. Therefore, through qRT-PCR, the most relevant collagen isoforms were checked in fibroblasts untreated or treated for 48 h with 20 IU/mL BoNT (see also Appendix A). No significant differences were reported for the *COL1A1*, *COL3A1*, *COL4A1*, *COL5A1*, *COL5A2*, *COL8A1* and *COL11A1* mRNA transcript levels in fibroblasts treated for 48 h in vitro with BoNT, nor in CP versus TD fibroblasts or in CP fibroblasts at different time points after in vivo BoNT injection (Figure 2A). In addition, the main collagen molecules, which are important in the polymerization of collagen fibers in skeletal muscle [66,67,68], were analyzed at the protein level by grouping the patients by GMFCS level and comparing them to TD samples in the same age range. Unfortunately, no GMFCS-III analyses are reported due to the low number of samples (n = 2). No statistical differences for COL1A1, COL3A1 and COL5A1 protein levels were reported after BoNT treatment in GMFCS-I CP versus TD fibroblasts (Figure 2B,C). The protein levels of COL1A were significantly decreased in CP fibroblasts from GMFCS-II patients compared to TD data. Still, no significant differences could be observed in the levels of COL3A1 or in the ratio between COL1A1 and COL3A1 (Figure 2D,E). For the longitudinal analysis, we introduced FIBRONECTIN 1 (FN1) as a niche component that can influence the regenerative capacity at later time points [69], together with the two main collagens (COL1A1 and COL3A1) as extracellular matrix constituents. No differences were reported in the levels of FN1, COL1A1 and COL3A1 in fibroblasts from children with GMFCS-I at t1 (=3 months) and t2 (=6 months) after in vivo BoNT administration (Figure 2F,G), while an increase at t1 (n = 2 patients) followed by lower levels comparable to those of TD children was observed 6 months after in vivo BoNT administration in CP children with GMFCS-II (Figure 2H,I). Interestingly, when GMFCS-I fibroblasts from the three different time points were still exposed to BoNT treatment in vitro (as an acute treatment; see Appendix A), a decrease in COL1A1 and an increase in COL3A1 were reported. This resulted in a reduction in the ratio between the two collagens (Appendix A). The ratio of COL1A1 to COL3A1 at t2 increased for GMFCS-II fibroblasts compared to the TD ratio (equally treated; Appendix A).

Taken together, these data confirm that BoNT administration in the muscle does not induce significant differences in the levels of the main collagens or FN1 over 6 months. Direct in vitro BoNT treatment of muscle-resident fibroblasts can induce a decrease in COL1A1 secretion of fibroblasts in CP children with GMFCS-II, as already hypothesized in a similar model [31].

### 3.4. Generation of Autologous hiPSC-Based Neuromuscular Junction Model

To further study the effect of BoNT on the lower MNs, especially in affecting their signal transmission to the muscle fibers, NMJs were generated in vitro using hiPSCs. Multinucleated myotubes were generated in less than 20 days from three different hiPSC lines of healthy donor origin [43]. HiPSCs expressing pluripotency markers such as NANOG (green) and OCT4 (red), as shown via IF (Figure 3A), were differentiated to multinucleated myotubes (syncytia; Figure 3B) expressing nuclear MYOD and cytoplasmatic MYHC. WB analysis confirmed the protein expression of OCT4 on day 0 of differentiation (M0) for all three lines. In a subsequent stage of differentiation (M3), no pluripotency markers were present, and high expression of MYHC, MYOD and ACTN2 proteins after complete differentiation was reported (M3; Figure 3C).

After neural differentiation, lower MNs from healthy donor hiPSCs did not show any expression of NANOG (green) or OCT4 (red; Figure 3D). Lower MN differentiation was confirmed by the IF results of NEFH (green) and choline acetyltransferase (CHAT; red) or ISLET1 (ISL1; green) and b3-tubulin (b3TUB; red; Figure 3E). The protein levels during the MN differentiation protocol were determined for pluripotency marker OCT4 and markers for early neural induction, such as NEST and PAX6 (Figure 3F), as well as for the NEFH marker for terminal differentiation (N4; Figure 3G).

Interestingly, autologous NMJs were generated from three different control cell lines after seeding hiPSC-derived myoblasts and MNs in microfluidic devices and inducing differentiation to myotubes (Figure 3H; higher magnification inset on the right). Thanks to the chemical attraction, neurofilaments were induced from the neural-dedicated compartment (Figure 3I, higher panel) to cross the microgrooves and enter into the myotube compartment (Figure 3I, lower panel). In the microfluidic chambers, many NMJs could be spotted by the overlap of NEFH (green) on myotubes (ACTN2; Figure 3J, left panels and insets on the right).

### 3.5. BoNT Treatment of Neuromuscular Junctions: Morphological and Molecular Effects

In order to directly assess the effect of BoNT on the target, namely NMJs connecting lower MNs and myotubes, we used bright-field (BF) images showing NMJs on microfluidic devices obtained by the juxtaposition of thin neurofilaments on myotubes (Figure 4A). These were more precisely highlighted by the colocalization of Bgt^+^ areas (bungarotoxin, red) on ACTN2^+^ myotubes (grey) colocalizing with NEFH^+^ neurites (green; Figure 4B, higher panels). After a 48 h treatment with 20 IU/mL BoNT, axonal swelling (white arrows) and axonal fragmentation (yellow arrows) were observed (Figure 4B, lower panels). These signs of stress were absent in the untreated compartment (Figure 4B, upper panels) but selectively present under BoNT-treated conditions (Figure 4B, lower panels). Axonal swelling (Figure 4C) and axonal fragments (Figure 4D) were significantly increased in BoNT-treated NMJs. Taken together, these data confirm that BoNT induces alterations that significantly affect the lower MNs.

In order to investigate the impact of BoNT at the molecular level, a time course and a quantification during 5 days of 20 IU/mL BoNT treatment of NMJ cocultures were performed (see Appendix A). On one side, a continuous decrease in the levels of SNAP25, the molecular target of BoNT (Appendix A), was already visible after 48 h and remained at lower levels during the entire 5 days of treatment. On the other side, 20 IU/mL BoNT treatment resulted in an increase in Atrogin1 protein levels, which was already visible after 24 h and maintained high levels during the following 4 days of treatment (Appendix A). These data show the cleavage of SNAP25 in the neural compartment and the increase in one of the ubiquitin ligases in the muscle compartment as a result of BoNT treatment.

### 3.6. Repetitive BoNT Treatment of Neuromuscular Junctions

One of the first-line treatments used to reduce spasticity in muscles from children with CP is BoNT administration, with repetitive administrations in time. To address the effect of several BoNT administrations on NMJs, cocultures of lower MNs and myotubes were repetitively stressed with in vitro BoNT treatments. The cocultures were treated for 48 h with 20 IU/mL BoNT, followed by 48 h recovery in normal medium and compared to untreated cocultures. This procedure was applied twice (see Appendix A). No statistical differences were reported for ACTN2 and MYHC after BoNT administration (Figure 5A,B), confirming complete differentiation of myotubes and no influence of BoNT treatment on those proteins. Therefore, besides the proteolysis mediated by proteasome and initiated by Atrogin1 activation, we investigated whether other pathways of induced proteolysis could contribute to muscle atrophy. The protein levels of three autophagy markers (BECL1, LC3B-I and LC3B-II) and p62 were tested. After 48 h and 96 h of treatment (BoNT 1× and 2×, respectively), the levels of BECL1, a protein that initiates the formation of autophagosomes, remained stable and similar to the control untreated levels (Untr 1× and 2×; Figure 5C,D). In addition, the levels of lipidated LC3B, which is a marker of newly formed autophagosomes, did not significantly differ over time (Figure 5C,D). Interestingly, the levels of p62, a cargo protein able to selectively tag proteins that need to be degraded via autophagosomes, were significantly higher after the second recovery period (BoNT + Rec ×2), confirming an accumulation of this tag protein that cannot be easily degraded as during BoNT treatment (Figure 5C,D). Autophagic involvement seemed to be limited to the neural component (or to both components if together), since SC-derived myotubes originated by myoblasts extracted from TD and treated with 20 IU/mL BoNT for 48 h did not show any increase in p62 levels (see Appendix A). This means that in muscle cells, autophagic degradation can easily counteract toxin-induced proteolysis. In line with these results, BoNT also does not cause any alteration of the levels of p62 or Atrogin1 in myotubes obtained from healthy hiPSCs (Appendix A), indicating that one BoNT treatment does not cause any accumulation of these two proteins in myotubes. The latter results highlight the importance of the neural compartment and its contribution for the increase in autophagic hyperactivation.

Taken together, these data confirm that besides the already described proteasome activation, autophagy contributes to proteolysis after a single BoNT administration. Interestingly, we demonstrated that autophagic hyperactivation is no longer efficient after two cycles of BoNT treatment, resulting in the accumulation of p62.

## 4. Discussion

The goal of this study was to clarify the direct effect of the first BoNT administration on the main cell types that play a role in muscle growth in young BoNT-naïve CP children. We focused on SC-derived myoblasts differentiated to myotubes and on fibroblasts promoting collagen deposition. The study was planned to explore possible alterations following BoNT treatment that could help to explain the growth delay seen in CP children after in vivo BoNT administration in comparison to age-matched TD children [24,25,26,27,28]. These effects were uniquely followed over time in the same BoNT-naïve CP children, with particular attention on the muscle cells extracted 3 (t1) and 6 months (t2) after the first in vivo BoNT administration. Furthermore, to examine the effect of single or repeated in vitro BoNT treatment, using hiPSC technology and deriving autologous myoblast and lower MNs, we were able to form NMJs capable of uniquely recapitulating the effect of BoNT on the neural component present in the muscle.

In response to our initial hypothesis, we demonstrated no effect of in vitro BoNT treatment on the differentiation of SC-derived myotubes from young BoNT-naïve children with CP in comparison to those derived from TD children in the same age range. Strikingly, comparison of the FI of cells extracted at t1 (= 3 months) and t2 (= 6 months) after the first in vivo BoNT administration showed a decrease in fusion abilities after 6 months, confirming long-term alterations of myoblasts after the first in vivo BoNT administration. Unexpectedly, no substantial modifications of the collagen synthesis in fibroblasts extracted from muscle microbiopsies derived from the same patients and TD children were reported in the current study. To achieve these aims, primary cells obtained from young BoNT-naïve patients with CP enrolled in a longitudinal study at different (short) time points after their first in vivo BoNT administration were collected to analyze the direct effect of BoNT and compared to the same cell types extracted from the same muscles of TD subjects in the same age range. Therefore, muscle specificity and the localization of the origin material were two features that were kept constant during the entire study. This could avoid muscle heterogeneity and other technical differences leading to variable interpretations of the results [70]. Finally, after double in vitro BoNT treatment of a novel hiPSC-derived NMJ model, we successfully defined the effect of BoNT on both muscle and neural compartments. An ongoing increase in an ubiquitin ligase of the ubiquitin-proteasome pathway and an impairment in p62 degradation from the autophagic–lysosomal pathway confirmed BoNT-induced proteolysis after repetitive treatments. This was clearly not limited to the muscle compartment but also possibly impaired in the vacuole trafficking at the neural level.

In the current analysis, in vitro BoNT treatment of SC-derived myoblasts gave rise to myotubes characterized by similar FI, independent of CP versus TD origin, clearly confirming no direct effect of BoNT on the ability of SC-derived myoblasts to form myotubes, as expected, and no intrinsic difference between CP and TD SC-derived myoblasts, which, similarly, do not differ due to in vitro BoNT treatment. Further in-depth investigations could better quantify specific contractile protein levels through WB analysis in order to confirm the results regarding the comparison between FIs at t0. Therefore, it is clear that there is no direct effect of BoNT on SC-derived myoblast differentiation, nor of the disease severity on specific cells; rather, the effect may be mediated by the niche, namely other cells in the in vivo environment of SCs, or by the interaction of these with molecules present in the same environment. The dose and concentration used for in vitro BoNT treatment were similar to clinical doses [24,25,35] in order to minimize possible toxic effects of BoNT but maximize the translational meaning of the assay. Moreover, this dose was chosen based on previous results on myoblasts and myotubes obtained from spastic quadriceps muscles of adult stroke patients [31].

Interestingly, a decreased FI was observed in CP-derived myotubes from myoblasts extracted 6 months after in vivo BoNT administration. This observation underlines a difference intrinsic to the cells (therefore, not due to the in vitro BoNT treatment) and is attributable to the in vivo BoNT administration, which affected the cells in the treated muscle extracted 6 months later. This is the first time that similar results have been reported in muscle stem cells from young CP patients and in the same children at different time points. These data are in line with what was previously reported in a macroscopic study, in which ultrasound examination of muscles treated with BoNT from young CP children showed a decreased volume for the MG [25], confirming the tendency of BoNT to impair treated muscles and raising concerns about the possibility of this neurotoxin delaying muscle growth in young children. Alterations of myoblast fusing ability could be responsible for the reduced FI reported in the present study in cells extracted 6 months after in vivo BoNT administration, although further characterization is required to confirm these speculations. These results could find support in the modulation of cell cycle genes induced after BoNT treatment in spastic quadriceps adult myoblasts [31]. A dedicated proliferation assay and a larger study group will help to draw final conclusions on these correlations and about the possibility of an epigenetic influence associated with BoNT [71,72].

Unexpectedly [5,9,10], the transcript levels of various analyzed collagens did not show major effects of in vitro BoNT treatment, not even on fibroblasts or at any time point after in vivo BoNT administration. Testing for extracellular matrix protein modulations, which are important for the formation of stable fibrils in the muscle, revealed no significant differences in COL5A1, COL3A1 or COL1A1 in the muscle of children with CP (GMFCS-I) at t0 after BoNT administration. COL5A1 has already been suggested as crucial in young adult mice because the depletion of this molecule induces anomalous cell cycle entry and differentiation, resulting in a decrease in the SC pool [73]. Further research can better elaborate this in human conditions; it would be useful to study the expression of COL5A1 in children with CP over time. However, a reduction in COL1A1 for GMFCS-II-derived cells from CP children attracted our attention. These data are in line with transcriptional data reported in spastic conditions of quadriceps fibroblasts from stroke patients [31] and may be related to collagen depletion, specifically for some muscles (like *soleus* or *MG*) in comparison to others (semitendinosus or gracilis), as previously published [74]. Several authors have reported a significant increase in collagen in different muscles (vastus lateralis or flexor carpi ulnaris) in spastic CP subjects that were older than our study group compared to TD peers [5,10]. Therefore, further studies including a larger sample size are warranted in order to better understand the role of age and degree of involvement in the pathology.

In the longitudinal analysis, no differences were reported for the levels of FN1, a protein influencing the regenerative capacity of SCs at longer time points, underlining no influence of BoNT and no alterations over time due to the progression of the pathology. Furthermore, the ratio between COL1A1 and COL3A1, which represents the rigidity and tensile strength over elasticity [66], goes back to TD levels at t2 after initially low levels, confirming no significant differences over time or due to BoNT administration, at least for this group of young children. Other studies in CP muscle undergoing tendon lengthening, i.e., at later time points in the progression of the pathology, have suggested that collagen could potentially be synthetized by fibroblasts, although based on data from different muscles, older children and different GMFCS levels [75,76]. Therefore, after these analyses, our initial hypothesis of an increased collagen deposition in CP fibroblasts after BoNT administration (or in vitro treatment) is not confirmed, and further molecular and histological analyses of muscles from older patients with CP could help to determine at which age collagen accumulation starts, influencing the presence of endomysial connective and fibrotic (scar) tissue. Another important conclusion at this point is that the spasticity of CP muscles, especially after BoNT administration, does not result from an increase in collagen content. Further investigation is required in order to better understand the nature and the main actors involved. Importantly, more attention should be paid to the neural component and the cytokines secreted in the case of injury [77,78].

An important advantage of our study is the possibility of directly observing the effect of BoNT on the neural target. For the first time, axonal swelling and axonal fragmentation were demonstrated in NMJs treated with a single dose of BoNT and compared to untreated NMJs. These observations, together with the decrease in SNAP25, which is the target of BoNT, are in agreement with those reported in another study aimed to compare the effect of different BoNT molecules. The study showed that 24 h exposure to low concentrations of different BoNT molecules was associated with SNAP25 cleavage in hiPSC-derived MNs, resulting in a decreased contraction of human immortalized myotubes [47]. For the first time, our study characterized a human autologous model that responds to a clinical single dose of BoNT with neural damage and recapitulates what is happening in the muscle, where impairment of contraction follows BoNT treatment. This model also shows that the muscle compartment responds to BoNT treatment by upregulating one of the muscle-specific ubiquitin ligases (Atrogin1), which is responsible for driving proteins destined for proteasome degradation and is active in many different pathologies characterized by muscle atrophy [79,80]. This points towards a similarity to the atrophy induced after BoNT, as demonstrated by studies of animals and a single dose administration in healthy human volunteers [6,15,49,50,51,81]. We did not see a decrease in the two main contractile proteins (MYHC and ACTN2). Future experiments should evaluate myogenic transcription factors such as MYOD and eukaryotic translation initiation factor 3 subunit f (eIF3-f [82,83]) or sarcomeric proteins such as DESMIN and VIMENTIN [84] to discover whether other targets of this ligase are decreased after BoNT administration.

Interestingly, for the first time, we showed that repetitive treatment of 48 h exposure to BoNT can induce engulfment of the lysosomal–autophagic system by causing a significant accumulation of p62, the cargo protein in charge of binding proteins destined for degradation. Our preliminary analyses show that this effect does not seem to involve the muscle compartment alone but was only detectable in coculture, leading to speculation regarding the neural compartment. However, further studies are needed to confirm the possibility of BoNT engulfing the neural component and NMJs, assuming that dysregulation in the release and reuptake of acetyl choline molecules could be linked to the engulfment of the lysosomal–autophagic system.

The main limitations of the current study are the small sample size and the intrinsic weakness of a single-center study. Further research with a larger study group would be preferable to obtain firm conclusions and to start deciphering the role of stem cells in altered muscle growth and contractility that our group and others have reported in recent muscle studies in children with CP. In addition, the in vitro model used in the current study could be further improved by recapitulating the condition in humans, particularly in young children with CP. Significant neurogenic muscle atrophy limited to a single injected gastrocnemius was reported following MRI evaluation of two healthy adults (14–19% reduction at 3 months; 27% at 6 months; 13–38% at 9 months; 12–22% at 12 months; [13]). In young children with CP, multiple reports have shown a loss of muscle volume between 1 and 3 months after the first BoNT injection, followed by almost complete recovery at 1 year [7,27]. A more recent study raised concerns about the frequency of injections, showing no complete recovery at 6 months [25], and suggested that longer intervals between reinjections are preferable. The possibility of counteracting atrophy with available drugs or accelerating recovery through the use of other treatments such as exercise or increased physiotherapy might fasten recovery after BoNT treatment [85,86].

In conclusion, for the first time, we clearly demonstrated that in vitro BoNT treatment of cells of young children with CP after their first in vivo BoNT administration was not harmful for the differentiation of SC-derived myotubes and fibroblast collagen deposition. Interestingly, a decrease in FI of myotubes extracted 6 months after in vivo BoNT administration supports the idea of an impaired stimulus for muscle growth after BoNT administration and fosters further investigation in order to understand whether this is a direct long-term effect on the cells or on the environment of the injected muscle. Thanks to the autologous NMJ model, we were able to highlight the effects of BoNT on the neural counterpart and shed light on the hyperactivation of the autophagic–lysosomal system in case of repetitive in vitro BoNT treatments. These findings promote further research about potential therapeutic candidates that may be able to counteract such a strong proteolysis and ameliorate the recovery strategy.

## Figures and Tables

**Figure 1 cells-12-02072-f001:**
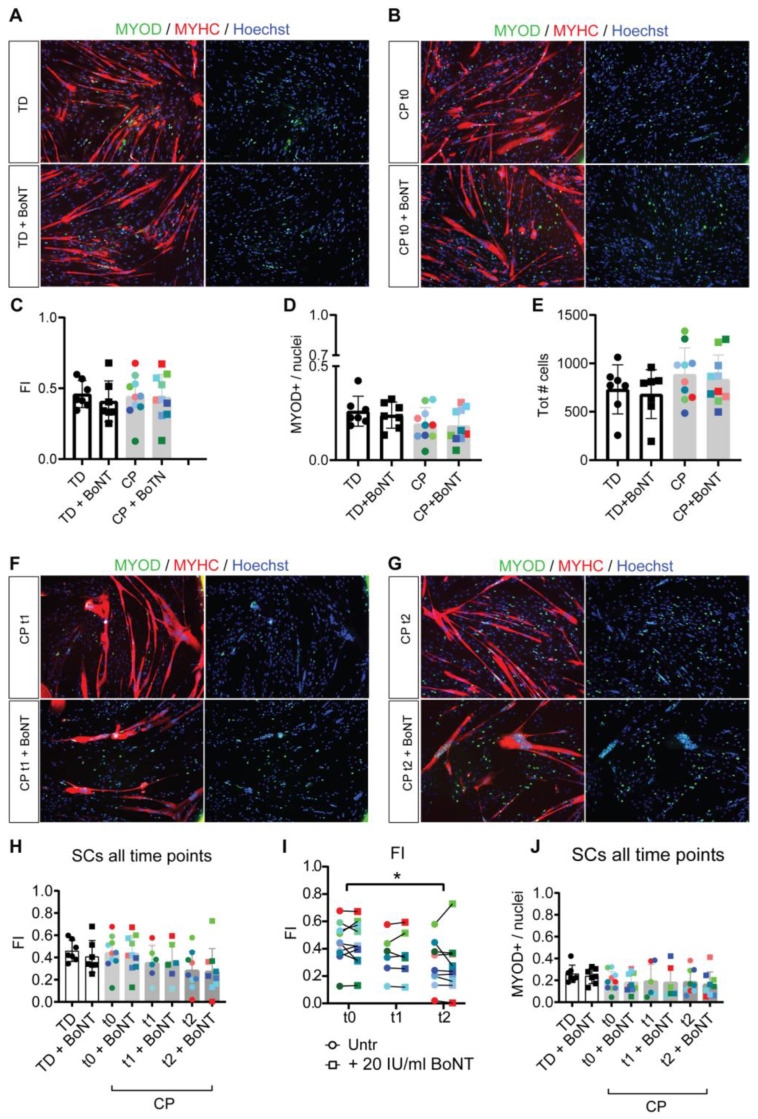
Myogenic differentiation of SC-derived myoblasts isolated from children with CP and TD in the same age range with or without in vitro BoNT treatment. (**A**) Immunostaining showing the expression of nuclear MYOD (green) and cytoplasmic MYHC in myotubes (red) from SC-derived myotubes originated from TD and (**B**) children with CP at time 0 (t0, before in vivo BoNT administration). These cells were untreated or treated with 20 IU/mL BoNT during 48 h of exposure before differentiation medium was applied for 6 days (for a visual representation, please see Appendix A). Nuclei were counterstained with Hoechst (blue). Scale bar = 500 μm. Quantification of (**C**) the fusion index (FI, ratio between the number of nuclei inside a myotube and the total number of nuclei appearing per field of view), (**D**) the MYOD fraction (ratio between the number of MYOD^+^ nuclei and the total number of nuclei) and (**E**) the total number of nuclei in TD children (n = 7) and children with CP (n = 10) recruited at t0 of the protocol. Immunostaining at (**F**) t1 (3 months after in vivo BoNT administration, n = 6) and (**G**) t2 (6 months after in vivo BoNT administration, n = 9) showing the expression of MYOD (green) in the nuclei and the localization of MYHC in the myotubes (red). Nuclei were stained with Hoechst (blue). Scale bar = 500 μm. Data represent independent experiments, and values are expressed as mean ± SD. Data were analyzed by one-way ANOVA, followed by Dunnet’s post hoc test. Quantification of (**H**,**I**) FI and (**J**) the MYOD fraction in TD children (n = 7) and children with CP recruited at t0 (n = 10), t1 (n = 6) and t2 (n = 9). Untreated samples are represented as round symbols, while treated samples (20 IU/mL BoNT) are represented by square symbols. The same color highlights cells from the same sample. Shades of green indicate GMFCS-I samples, of blue indicate GMFCS-II samples and of red indicate GMFCS-III samples. Data are representative of independent experiments, and values are reported as mean ± SD. Data were analyzed by one-way ANOVA followed by a Tukey’s post hoc test or two-way ANOVA followed by Dunnet’ post hoc test: * *p* < 0.05. CP t2 vs. CP t0.

**Figure 2 cells-12-02072-f002:**
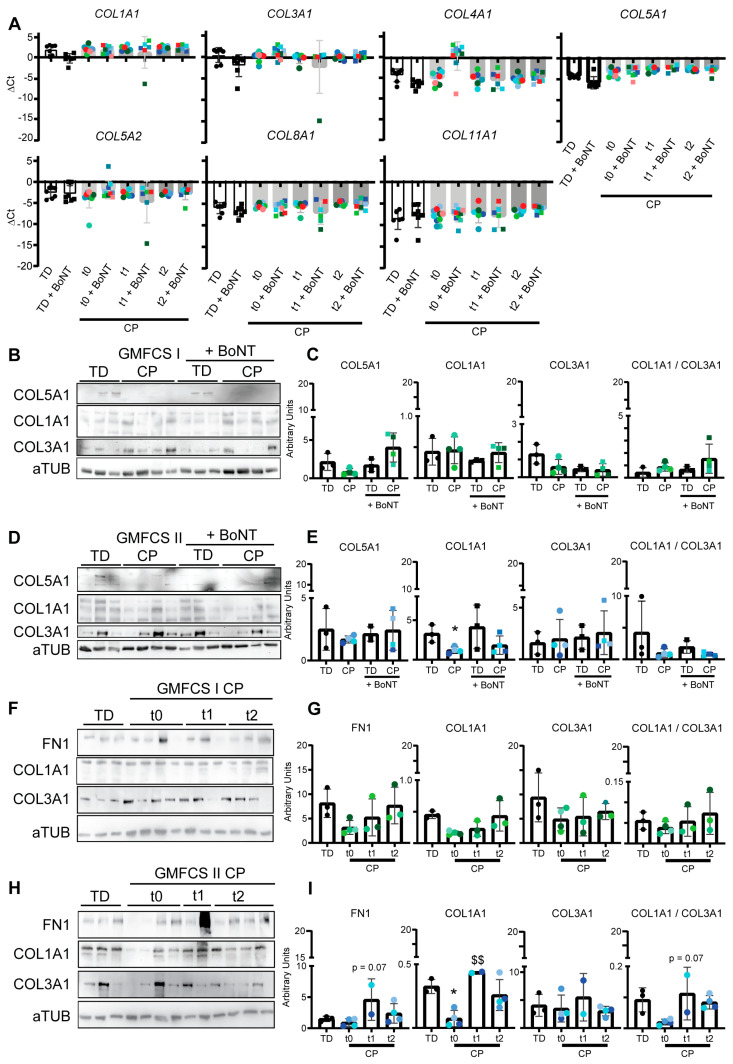
Extracellular matrix protein deposition in BoNT-treated muscle-derived fibroblast. (**A**) Gene expression profiles of *COL1A1*, *COL3A1*, *COL4A1*, *COL5A1*, *COL5A2*, *COL8A1* and *COL11A1* expressed by fibroblasts extracted from the *MG* muscle in CP at t0 (immediately before in vivo BoNT administration, n = 10), CP at t1 (3 months after in vivo BoNT administration, n = 6), CP at t2 (6 months after in vivo BoNT administration, n = 9) and in TD children in the same age range (n = 7) exposed to 20 IU/mL BoNT for 48 h (for a visual representation, see also Appendix A). Each data point represents individual values expressed as DCt and normalized for the housekeeping gene (*bACT*). Values are reported as mean ± SEM. Data were analyzed by two-way ANOVA with Tukey post hoc test. (**B**) WB and (**C**) protein level quantification for COL5A1, COL1A1 and COL3A1 normalized for aTUB expressed by fibroblasts extracted from TD and GMFCS-I CP samples untreated or treated with 20 IU/mL BoNT for 48 h. (**D**) WB and (**E**) protein level quantification for COL5A1, COL1A1 and COL3A1 normalized for aTUB expressed by fibroblasts extracted from TD and GMFCS-II CP samples untreated or treated with 20 IU/mL BoNT for 48 h. The same color highlights cells from the same sample. Different shades of green indicate GMFCS-I samples, of blue indicate GMFCS-II samples. Data were obtained from independent experiments, and values are reported as mean ± SD. Data were analyzed via one-way ANOVA with a Tukey post hoc test: * *p* < 0.05. CP vs. TD. (**F**) WB analysis and (**G**) protein level quantification for FN1, COL1A1 and COL3A1 normalized for aTUB expressed by untreated fibroblasts extracted from TD and GMFCS-I CP samples at t0 (immediately before in vivo BoNT administration), t1 (3 months after in vivo BoNT administration) or t2 (6 months after in vivo BoNT administration). (**H**) WB and (**I**) protein-level quantification for FN1, COL1A1 and COL3A1 normalized for aTUB expressed by untreated fibroblasts extracted from TD and GMFCS-II CP samples at t0 (immediately before in vivo BoNT administration), t1 (3 months after in vivo BoNT administration) or t2 (6 months after in vivo BoNT administration). Untreated samples are indicated by round symbols, while treated samples are represented by square symbols. The same color highlights cells from the same sample. Different shades of green indicate GMFCS-I samples, of blue represent GMFCS-II samples. Data were obtained from independent experiments, and values are reported as mean ± SD. Data were analyzed via one-way ANOVA with a Tukey post hoc test: * *p* < 0.05 for CP vs. TD; $$ *p* < 0.01 for CP t1 vs. CP t0.

**Figure 3 cells-12-02072-f003:**
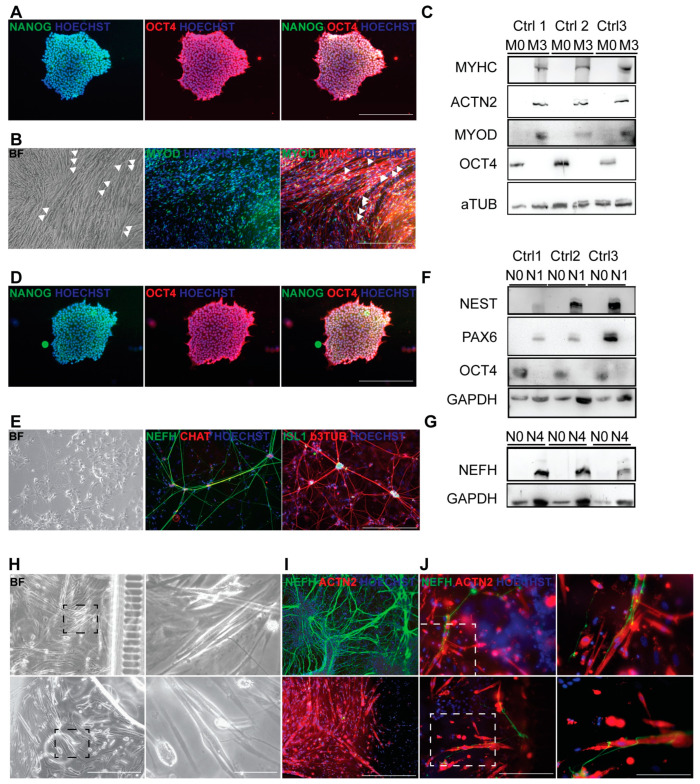
Characterization of hiPSC-based autologous NMJ coculture model of myotube and lower MN differentiation. (**A**) Immunostaining of a representative hiPSC colony expressing pluripotency markers NANOG (left panel), OCT4 (middle panel) and merged expression (right panel). (**B**) Bright-field image (left panel) and immunostaining showing myotubes (some of which are highlighted by the white triangles) with the expression of nuclear marker MYOD (green, middle panel) and cytoplasmatic marker MYHC (red, right panel). Nuclei were stained by Hoechst (blue). Scale bar = 500 μm. (**C**) WB of all three hiPSC lines used in the study showing the protein expression of OCT4 before induction of myogenic differentiation (M0) or MYHC, ACTN2 and MYOD at the end of myogenic differentiation (M3) normalized according to the levels of aTUB. (**D**) Immunostaining of a representative hiPSC colony expressing pluripotency markers NANOG (left panel), OCT4 (middle panel) and merged expression (right panel). (**E**) Bright-field image (left panel) and immunostaining showing lower MNs, as highlighted by the expression of NEFH (green, middle panel) and CHAT (red, middle panel) or by ISL1 (green, right panel) and b3TUB (red, right panel). Nuclei were stained by Hoechst (blue). Scale bar = 500 μm. (**F**) WB analysis of all three hiPSC lines used in the study, for the protein expression of OCT4, a pluripotency marker before induction of neural differentiation (N0), or PAX6 and NEST, in the first step of neural induction (N1) normalized by GAPDH or (**G**) the NEFH expression at the end of MN differentiation (N4) normalized by the levels of GAPDH. (**H**) Bright-field image (left panel) of microfluidic devices (left panels) highlighting the connections of lower MNs and myotubes (right panels). Scale bar = 500 μm for panels on the left, while scale bar = 50 μm for insets on the right. (**I**) Immunostaining of the two compartments present in the microfluidic devices, namely the lower MN compartment (NEFH; green, upper image) and the myotube compartment (MYHC; red, lower image). Nuclei were stained by Hoechst (blue). Scale bar = 500 μm. (**J**) Contact points between ACTN2^+^ myotubes (red) and NEFH^+^ filaments (green) representative for all untreated hiPSC lines are highlighted by white boxes in the panels on the left and enlarged in the insets on the right. Nuclei were stained with Hoechst (blue). Scale bar = 250 μm for panels on the left, while scale bar = 50 μm for insets on the right.

**Figure 4 cells-12-02072-f004:**
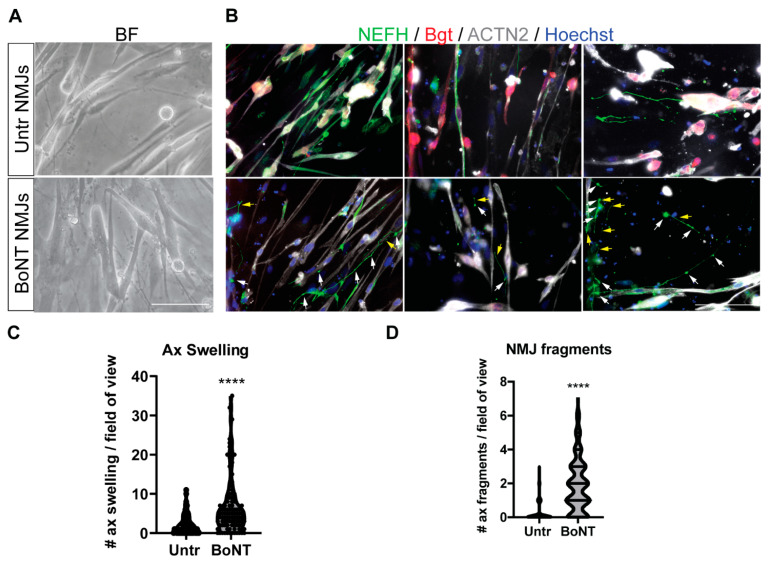
BoNT-induced morphological alterations in autologous NMJs. (**A**) Bright-field images (left panels) and (**B**) immunostaining of autologous NMJs highlighted by the colocalization between α-Bgt^+^ areas (red) binding to AChRs on ACTN2^+^ myotubes (grey) and NEFH^+^ filaments (green). Nuclei were counterstained with Hoechst (blue). Examples of axonal swelling (white arrows) and axonal fragmentation (yellow arrows) are reported. Scale bar = 50 μm. (**C**) Quantification of the amount of axonal swelling/picture and (**D**) of NMJ fragments per picture. Data were obtained from independent experiments, and values are reported as mean ± SD. Data were analyzed by *t*-test. **** *p* < 0.0001 for BoNT-treated vs. untreated NMJs.

**Figure 5 cells-12-02072-f005:**
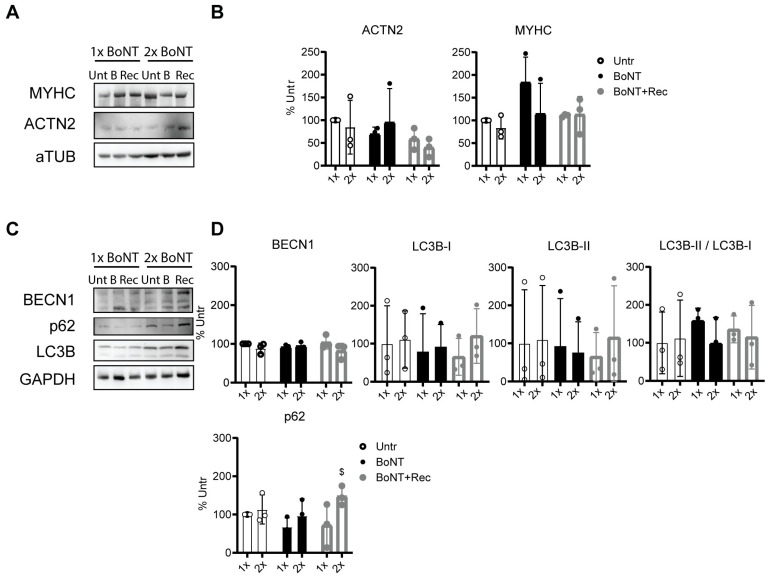
Repetitive BoNT treatments of NMJs engulf lysosomal–autophagic degradation. (**A**) WB and (**B**) protein-level quantification for ACTN2 and MYHC normalized for aTUB after no BoNT treatment (1×, Unt), 48 h treatment with 20 IU/mL BoNT (1×, BoNT) or 48 h treatment with 20 IU/mL BoNT followed by 48 h recovery in normal medium (1×, BoNT + Rec), as well after 96 h untreated (2×, Unt) or characterized by two treatments with 20 IU/mL BoNT (2×, BoNT) for 48 h or characterized by a cycle of 20 IU/mL BoNT treatment for 48 h followed by 48 h recovery in normal medium (2×, BoNT + Rec) repeated twice. Data were obtained from three independent experiments, and values are expressed as mean ± SD. Data were analyzed via two-way ANOVA with a Sidak post hoc test. (**C**) WB and (**D**) protein-level quantification for BECN1, LC3B-I and LC3B-II and the ratio between the two forms, p62 normalized for GAPDH and expressed by 48 h untreated (1×, Unt), 20 IU/mL BoNT-treated (1×, BoNT) or 20 IU/mL BoNT-treated followed by 48 h recovery in normal medium (1×, BoNT + Rec), as well after 96 h untreated (2×, Unt) or characterized by two 48 h treatments with 20 IU/mL BoNT (2×, BoNT) or characterized by a cycle of 20 IU/mL BoNT treatment for 48 h followed by 48 h recovery in normal medium repeated twice (2×, BoNT + Rec). Data are representative of three independent experiments, and values are reported as mean ± SD. Data were analyzed via two-way ANOVA with a Sidak post hoc test. $ *p* < 0.05 for 1× BoNT + Rec vs. 2× BoNT + Rec.

**Table 1 cells-12-02072-t001:** List of analyzed FACS, immunofluorescent (IF) staining and Western blot (WB) antibodies.

Protein	Antibody Name (#Catalog Number)	Provider	FACS	IF	WB
ACTN2	Anti-SARCOMERIC ALPHA ACTININ (EA-53) (Mouse monoclonal) (#ab9465)	Abcam, Cambridge, UK		1:200	1:500
ALKALINE PHOSPATASE-PE	Human/Mouse/Rat Alkaline Phosphatase PE MAb (Cl B4-78), Mouse IgG1 (FAB1448P-100)	R&D systems, Minneapolis, MN, USA	1.2 µL/10^6^ cells		
ATROGIN1	Atrogin-1 (Rabbit polyclonal) (AP 2041)	ECM Bioscences, Aurora, CO, USA			1:200
BECLIN-1	Anti-Beclin1-BH3 Domain (Rabbit polyclonal) (SAB1306537)	Merck, Darmstadt, Germany			1:1000
CD56-APC	APC anti-human CD56 (NCAM) Mouse IgG1 (clone MEM-188) (#304610)	BioLegend, San Diego, CA, USA	0.1 µL/10^6^ cells		
COL1A1	COL1A1 (E8F4L) XP (Rabbit monoclonal Ab) (#72026)	Cell Signaling, Danvers, MA, USA			1:1000
COL3A1	COL3A1 (E8D7R) XP (Rabbit monoclonal Ab) (#63034)	Cell Signaling, Danvers, MA, USA			1:500
COL5A1	COL5A1 (E6U9W) (Rabbit monoclonal Ab) (#86903)	Cell Signaling, Danvers, MA, USA			1:500
FN1	Fibronectin1 (E5H6X) (Rabbit monoclonal Ab) (#26836)	Cell Signaling, Danvers, MA, USA			1:500
GAPDH	Anti-GAPDH (Rabbit polyclonal) (G9545)	Merck, Darmstadt, Germany			1:1000
ISL1	Recombinant Anti-ISLET 1 (EP4182)—Neural Stem Cell Marker (Rabbit monoclonal) (#ab109517)	Abcam, Cambridge, UK		1:500	1:60
LC3B	Anti-LC3B (Rabbit polyclonal) (L7543)	Merck, Darmstadt, Germany			1:1000
MF20	MYHC1 (MF20) (Mouse monoclonal)	Develop Studies Hybridoma Bank (DSHB), IOWA, USA		1:300	1:3
MYOD	MYOD1 (D8G3) XP Rabbit mAb (Rabbit monoclonal) (#13812)	Cell Signaling, Danvers, MA, USA		1:300	1:1000
NANOG	NANOG (Rabbit polyclonal) (#PA1-097)	Thermo Fisher Scientific, Waltham, MA, USA		1:200	
NEFH	Anti-NEUROFILAMENT HEAVY POLYPEPTIDE antibody (Rabbit polyclonal) (#ab8135)	Abcam, Cambridge, UK		1:1000	1:100
NEST	Purified anti-NESTIN (10C2) (Mouse monoclonal) (#656801)	BioLegend, San Diego, CA, USA			1:250
PAX6	PAX6 (D3A9V) XP Rabbit mAb (Rabbit monoclonal) (#60433)	Cell Signaling, Danvers, MA, USA			1:150
PDGFRa-APC	APC anti-human CD140a (PDGFRα) Mouse IgG1 (clone 16A1) (#323512)	BioLegend, San Diego, CA, USA	0.1 µL/10^6^ cells		
p62	Anti-p62 (Mouse monoclonal)	BD Bioscences, NJ, USA			1:9000
SNAP25	Anti-SNAP-25 (Rabbit polyclonal) (S9684)	Merck, Darmstadt, Germany			1:200
TUBA4A	Anti-a-TUBULIN (B-5-1-2) (Mouse monoclonal) (#T6074)	Merck, Darmstadt, Germany			1:1000
TUBB3	Anti-beta III TUBULIN (2G10)—Neuronal Marker (Mouse monoclonal) (#ab78078)	Abcam, Cambridge, UK		1:500	1:300

**Table 2 cells-12-02072-t002:** Primer list for qRT-PCR.

Gene	Primer Direction	Primer Sequence (5′ > 3′)
*COL1A1*	Forward	CCTGGATGCCATCAAAGTCT
	Reverse	TCTTGTCCTTGGGGTTCTTG
*COL3A1*	Forward	AAGAATTTGGTGTGGACGTTG
	Reverse	TTTTGTCGGTCACTTGCACT
*COL4A1*	Forward	CCAGGATTTCAAGGTCCAAA
	Reverse	CTC CCCTTTGATGATGTCGT
*COL5A1*	Forward	CCTGACCCTGGACAGTGAAG
	Reverse	GGCTCCTTCCCTCTGTTCTC
*COL5A2*	Forward	TCAAAAGAAGCCTCCCAGAA
	Reverse	TCTAAGTCATTTGCCCCTTTG
*COL8A1*	Forward	ACCACCCCAGGGAGAGTATC
	Reverse	AATGCAGGCATCTCATAGGC
*COL11A1*	Forward	GCATTTTGATGCTTTATTCAAGG
	Reverse	CACACATTTCCCTGTCCAAA
*ACTB*	Forward	GGACCTGACTGACTACCTCAT
	Reverse	CGTAGCACAGCTTCTCCTTAAT

**Table 3 cells-12-02072-t003:** Demographic characteristics, anthropometric data, clinical symptoms and treatment characteristics for the study participants.

		TD (n = 7)	CP t0 (n = 10) GMFCS I n = 4 GMFCS II n = 4 GMFCS III n = 2	CP t1 (n = 6) GMFCS I n = 3 GMFCS II n = 2 GMFCS III n = 1	CP t2 (n = 9) GMFCS I n = 3 GMFCS II n = 4 GMFCS III n = 2
Age of first biopsy (year)	Mean (SD)	6.19 (1.92)	4.11 (1.66)	5.2 (1.7)	6.1 (2.0)
	Range	4–7	2–7	3.1–7.5	3.7–8.9
Gender (m-f)	n	5–2	5–5		
Body mass (kg)	Mean (SD)	22.18 (7.06)	15.60 (6.52)	17.4 (2.8)	19.5 (4.6)
Height (cm)	Mean (SD)	118.36 (14.70)	100.15 (14.62)	104.5 (10.4)	113.6 (15.5)
Involvement (unilat-bilat.)	n	-	5–5		
Physiotherapy (min/week)	Mean (SD)	-	139.7 (60.1)	123 (60.4)	158.6 (56.7)

In this study, two subject groups were included: typically developing (TD) children and cerebral palsy (CP) patients. Children with CP were further subdivided based on Gross Motor Function Classification System (GMFCS) levels and topographic classification (unilateral or bilateral involvement). The data refer to the number of participants, mean and standard deviation (SD).

## Data Availability

Not applicable.

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
