# Peer review of "Botulinum Toxin Treatment of Adult Muscle Stem Cells from Children with Cerebral Palsy and hiPSC-Derived Neuromuscular Junctions"

_cells, 2023, doi:10.3390/cells12162072_

Round 1
Reviewer 1 Report
Comments on Costamagna et al, manuscript submitted Cells cells-2482679
“Botulinum Toxin Treatment of Adult Muscle Stem Cells from Children with Cerebral Palsy and hiPSC-derived Neuro Muscular Junctions”
In this manuscript, the authors investigate the impact of Botulinum Neurotoxin Type A (BoNT) on muscle biopsies from Cerebral Palsy (CP) patients taken at different timepoints and a human-derived iPS neuromuscular junction (NMJ) model. They conclude that an acute treatment of BoNT does not impair satellite cell (SC)-derived myotube differentiation or fibroblast collagen production. However, they observed that SC-derived myoblasts obtained 6 months after BoNT treatment showed a significant decrease in fusion abilities, suggesting possible long-term alteration of SCs following the first BoNT injection. In addition, they show that repeated BoNT treatments in the NMJ model result in a defective autophagic-lysosomal system.
The manuscript is accompanied by clear figures that are displayed in a logical order. The manuscript would benefit from some significant English editing. The study describes a precious dataset of patient-derived biopsy material. Therefore, the reviewer thinks this manuscript is a useful addition to the field. However, there are several major caveats. The most important one being that there is no control group of CP material lacking BoNT treatment. This way it is not possible to conclude whether the observed post BoNT effects are BoNT related or are already the result of having CP. The conclusions related to this should therefore be revised. In addition, the conclusion that the first in vivo BoNT administration is not harmful for SC-derived myotube differentiation seems somewhat premature though, considering the myoblasts show a significant decrease in fusion ability 6 months post administration suggesting an impaired stimulus for muscle growth. This section should be revised. The reviewer recommend major revisions. Given this reviewers expertise lies mostly with the NMJ and the models used, I advise to editor to also include a reviewer who is familiar with the cerebral palsy and BoNT field to provide sufficient feedback on the impact and context represented in this manuscript given that topic.
Specific feedback is given below, broken down into major and minor comments.
Major comments
1. Did the authors include a CP group with no BoNT (mock) treatment as apparently these patients are already known to suffer from SC disfunction (written in intro as reference)? So how do we know the SC effects observed after BoNT treatment are related to the treatment and not the disease itself? In the materials and methods section it’s not clear to the reviewer whether the TD group has no CP (the biopsy section later on may suggest it to be healthy donors but more clarity in M&M would help).
2. The introduction is rather long and it’s difficult to extract the aims of the study. There are several hypothesis mentioned and also a mixture of aims on fibroblasts, SCs etc. Readability of the manuscript would improve if the authors can bring their hypothesis and objectives across in a more structured and focused manner.
3. Did the authors observe any differences between results of SC etc based on disease severity in the patients?
4. The dosing differed between patient. Did the authors observe any differences between the dosing and later cellular function? Was there any correlation/dosing bias?
5. It is unclear to the reviewer why a healthy childrens population (TD) (and hospital ethical committee) would allow such biopsies etc. were these children in the hospital for other reasons? Some more clarification on this control group would be helpful. This info is later on given in biopsy section I believe but feels out of place and doesn’t discuss ethics.
6. Table 2 (line 234) in this manuscript does not correspond with the table that has been described in text. From the context of the manuscript the reviewer believes this is meant to be supplemental table 2. The same can be said for table 3 (427).
7. Regarding the conclusion drawn in line 492-495: The observation is made that BoNT injection results in decreased FI value over time and a tendency to decrease the total number of MYOD+SC-derived myoblasts after myogenic differentiation (referring to figure 1H and 1I respectively). However, looking at these graphs the differences between timepoint do not appear significant. Considering the sample size the reviewer wonders if it is appropriate to draw this conclusion based on the data provided. In line 458 the authors refer to the significantly lower FI overtime as well, referring to figure s1B. This seems like an important finding and should probably be included in the main figure. That said it is strange that these differences are not at all seen at the different timepoints in figure 1H.
8. In general the numbers per time point in figure 2 are really quite low and therefore it’s hard to make any final conclusions as statistics on such small sample sets can be questions when it comes to variable human nature. It’s also very difficult to see the squares vs the circles with the signifcancy lines running over it and symbols being on top of each other.
9. There seems to be a disparity between figures 2F and 2G (526). In the WB (2F) FN1 shows a very strong band in the FN1 staining at t0 (CP t0 lane 6 of WB). However, this point does not seem to be represented in the quantification in 2G.
10. Figure 2 (526). It is unclear what GMFCS classification the datasets (B and C; D and E etc.) belong to. The reviewer recommends including the information in the graphs to avoid confusion.
11. Figure 4 BTX staining looks very weird. Do the authors have an explanation for this? It does not look like anything that has previously been reported and the reviewer wonders whether these are truly AChR clusters or cell death related staining? Also from panel A to which figure does the brightfield relate to? These seem random pictures from different fields put together but the bright field does not represent any of them. Consider presenting this data in a more structure manner.
12. Figure 4E please show datapoints from individual experiments to give a sense for the variation among experiments. And N=1 does not suffice.
13. 623-624: It is unclear how axonal swelling and NMJ fragmentation was quantified. This is also not explained in the results section of the paper. Please provide information on how both of these were quantified. In addition, the reviewer wonders if the pictures chosen in figure 4A are fully representative of the striking difference reported in 4B and 4C. Based on visual cues alone the differences (to an untrained eye) look minimal.
14. 670: Increased p62 following repeated BoNT treatments is an interesting observation as it indicates that the autophagic-lysosomal system may be affected as a result of it. Therefore it may be of interest to show this data as part of the main figure rather than in the supplemental. Showing the data will give more strength to the conclusion drawn in this section.
Minor comments
1. Line 26: Add ‘The’ before Goal
2. Line 46: to NMJ co-cultures should be in NMJ co-cultures. There are some other places in text where the wrong preposition word is used (e.g. line 100, 103 etc.). There are also other minor grammatical errors such as misplaced commas or wrong verb tenses. The reviewer suggests that the manuscript will be edited thoroughly before publication
3. Line 75: Synapse instead of synapsis
4. Line 124 does not read well
5. What’s the sex of the donor iPSC lines?
6. paragraph 1 of material and methods sections can be removed as it reads like a content description which imo is not needed for an M&M section.
7. Line 148: Finishing is the wrong verb tense
8. Line 177: Unclear why gender and involvement (unilat- bilat.) was not included for CP t1 and t2 groups.
9. Line 224-229 is a very long run-on sentence that is hard to read. Consider rephrasing or breaking up into smaller sentences.
10. Line 329-311 capitalization is not the formal standard to write human protein names
11. Line 440: remove at. Consider rephrasing as currently the title is unclear.
12. Line 441, this sentence is extremely long and for readability should be revised
13. The colors chosen to represent different GMFS-scores in the bar graphs (e.g. figure 1C) are quite close together in hue. This will make it very hard to distinguish between the scores when looking at it in grayscale or viewed by a colorblind individual. Consider changing this to improve accessibility.
14. Line 507: age range instead of range of age
15. line 555 dollar signs?
16. In figure 2 (526) there are quite a few bands present for different collagens, in particular COL1A1 (see figure 2B, 2D, 2F, and 2H). What band was analyzed and how did the authors determine they were looking at the right band?
17. In the western blots throughout the paper the levels of loading control vary greatly (for example in figure 3G). While signal can be (and has been) adjusted to the amount of loading control in that lane, loading equal values on the gel is likely to give more representative blots.
18. The paper reads a bit schizophrenic as the part on iPSC-based NMJs reads much better than many other parts of the manuscript. The entire paper should be revised by the person who corrected this part of the manuscript.
19. Figure 4A (635): in the right most picture of the BoNT treatment NMJs there appears to be a white arrow that does not point at anything. As the white arrows have no tails it is quite difficult to determine in which direction they point. Consider changing this for improved clarity.
20. Line 734-738: This sentence requires editing. As is it is very long and hard to follow.
21. Line 767-768: collagens instead of collagens
22. Line 769: ‘At all (short) time points’ is phrased awkwardly and the addition of short feel obsolete. ‘At any timepoints’ flows nicer.
23. Line 784-785: Last part of sentence ‘that just further…. understand’ requires rephrasing
24. Line 835: ‘The main limitation here of the current study….’ is not grammatically correct. Reviewer suggests: ‘The main limitation of the current study is…’
needs significant English editing
Reviewer 2 Report
A well thought out paper. I do not have any "content" revision suggestions. A limitation of this study is that the sample sizes were relatively small. Therefore, further studies are warranted.
Reviewer 3 Report
Dear Authors,
In my opinion, the topic of your study is intriguing, given that in children affected by cerebral palsy botulin toxin treatment is commonly used, considering that it has shown to decrease muscle tone, ameliorate mobility and muscle function, ultimately determining gait improvement. However, some risks regarding such treatment have been highlighted, especially detrimental effects of BoNT on muscle growth and contractility. The ability to better understand such effects of botulin toxin treatment is an important resource considering the importance of this treatment in children with cerebral palsy.
However, I have concerns about the methodological implant of your study and some issues should be addressed to improve the paper.
Major Reviews
INTRODUCTION: This section should be improved highlighting the importance of botulinum toxin in a multidimensional approach aiming at improving quality of life of patients and of their caregivers. According to this, you should cite the following references:
· Lippi L, de Sire A, Folli A, et al. Multidimensional Effectiveness of Botulinum Toxin in Neuropathic Pain: A Systematic Review of Randomized Clinical Trials. Toxins (Basel). 2022;14(5):308. Published 2022 Apr 27. doi:10.3390/toxins14050308
· Kaya Keles CS, et al Botulinum Toxin Intervention in Cerebral Palsy-Induced Spasticity Management: Projected and Contradictory Effects on Skeletal Muscles. Toxins (Basel). 2022 Nov 8;14(11):772. doi: 10.3390/toxins14110772. PMID: 36356022; PMCID: PMC9692445.
· Kim H, et al Is botulinum toxin type A more effective and safer than other treatments for the management of lower limb spasticity in children with cerebral palsy? A Cochrane Review summary with commentary. NeuroRehabilitation. 2021;49(1):161-164. doi: 10.3233/NRE-218003. PMID: 34366300.
METHODS: The characteristics of the study participants have to be described in the “Results” section and not in the “Methods” section.
METHODS: Please specify who performed statistical analysis.
METHODS: The sample size calculation should be provided.
RESULTS: Please, report the number of individuals at each stage of study (numbers of potentially eligible, examined for eligibility, confirmed eligible, included in the study) and consider to use a flow diagram.
RESULTS: A Limitation section should be largely improved, including the intrinsic limitations of a single-center study with a small sample.
Minor Reviews
ABSTARCT: In accord to authors guidelines, the abstract should be 200 words maximum.
ABSTRACT: Line 46, please, correct the space issues: replace “co-cultures.in” with “co.cultures. In”.
WHOLE MANUSCRIPT: Line 143, please correct “and/ or” with “and/or”
Few typos should be addressed
Round 2
Reviewer 3 Report
Dear Authors,
in my opinion, the manuscript is interesting, and the results are intriguing.
You have significantly improved the paper during the revision process.
Therefore, in my opinion, the paper is now suitable for publication in this Journal.
Best regards